# The Relationship between the Second-Generation Antipsychotics Efficacy and the Traditional Chinese Medicine Body Constitutions in Patients with Schizophrenia

**DOI:** 10.3390/healthcare9111480

**Published:** 2021-10-31

**Authors:** Tzu-Pei Yeh, Li-Chi Huang, Yu-Fen Chen, Jui-Fen Cheng

**Affiliations:** 1School of Nursing, China Medical University, Taichung 406040, Taiwan; tzupeiyeh@mail.cmu.edu.tw (T.-P.Y.); lichi@mail.cmu.edu.tw (L.-C.H.); 2Department of Nursing, China Medical University Hospital, Taichung 404332, Taiwan; 3Department of Nursing, Taichung Veterans General Hospital, Taichung 40705, Taiwan; yfchen@vghtc.gov.tw

**Keywords:** schizophrenia, second-generation antipsychotics, traditional Chinese medicine, body constitutions

## Abstract

Background: Schizophrenia requires lifelong treatment; Second-generation Antipsychotics (SGAs) have become the most prescribed medication for schizophrenia patients. The efficacy of various SGAs treatment may differ in schizophrenia patients with various traditional Chinese medicine (TCM) body constitution (BC) types. Method: This study applied a longitudinal quantitative research design, where a total of 66 participants were recruited. The Positive and Negative Symptom Scale (PANSS) and the Clinical Global Impression (CGI) score were used to evaluate patients’ psychopathology status in hospitalization, and body constitution questionnaires were conducted by face-to-face interviews in the 1st, 3rd, and 6th week of hospitalization. Results: More than 60% of schizophrenia patients who were treated with SGAs were classified to have unbalanced BC types including Yin-Xu, Yang-Xu and Stasis. Generalized estimating equation analysis revealed significant time effects in CGI and PANSS score improvements in both unbalanced and gentleness (balance) BC types, but no significant changes in the group and group-time interaction in the CGI and PANSS scores in different BC type groups. Conclusions: Schizophrenia patients under SGAs treatment had a higher proportion of unbalanced BC types which may lead to poorer physical or mental statuses, such as overweight problems. Health care providers could apply interventions according to patients’ BC types for disease prevention.

## 1. Introduction

Schizophrenia is a chronic and severe mental disorder which affects various functions in daily lives. The most influenced impact of schizophrenia on social function is usually long lasting and the patients require lifelong treatment. Second-generation antipsychotics (SGAs) have become the most commonly prescribed drugs for schizophrenia patients worldwide, including Taiwan [1]. Although previous studies have demonstrated several advantages of SGAs compared to first-generation antipsychotics (FGAs), such as lower re-administration rates, shorter hospitalization duration, less adverse effects, less dosage of anticholinergic agents, and better medication adherence [2,3,4,5,6]; however, metabolic side effects among SGAs users have become a challenge for an aging society with the considerable efficacy and safety in antipsychotics treatments [7]. Medicine selection in schizophrenia treatment has been controversial as a clinical dilemma; the appropriate medication may request more individualized medication plans and consideration [2,4].

Individualized medicine with more specific treatments are the most important advances in current 21st century medicine guidelines [8], and it is mainly established based on genomics personal information. However, these genomics-based treatments are quite expensive; moreover, some challenges must be addressed before these treatments are applied broadly, such as ethical, security, privacy, and legal issues in genetic tests [9]. Traditional Chinese medicine (TCM) is an ancient medical system with a unique cultural background. TCM plays an important role in health maintenance for people not only in Chinese society but also globally nowadays [10,11], and it has been accepted as a major or combination complementary and alternative treatment approach in Western countries [12]. TCM is considered as one of the mainstream medicines in Taiwan, and increasing numbers of patients choose to be treated by using both TCM and Western medicine simultaneously [13,14]. The body constitution theory derived from the Canon of Internal Medicine or Treatise on Febrile and Miscellaneous Diseases of Ancient China, and it is an important part of the theoretical foundation in TCM [15]. Four principal factors influence body constitution (BC) types: heredity (genetic composition), ontogeny (stage of human development), psychological condition, and environment [12,16]. Body constitution is related to an individual’s vulnerability and the progression of disease. TCM treatments are often highly individualized and the first step is assessing and adjusting the individual’s body constitution [15,17,18,19]. This point of view highlights the genetic influence of disease vulnerability in Western medicine, which is similar to the concept of personalized medicine [19].

In TCM, a healthy human possesses balanced Yin and Yang dynamics; this healthy status may be presented through an individual’s BC type [20]. Yin could be referred to materials which drive and keep physiological functions, such as blood and fluid. Yang could be seen as the energy for body functioning, for instance, for heart beats, muscle movement or digestion. When Yang and Yin keep balance, the body function works smoothly [21]. In some circumstances, an individual’s behavior or pathologic factors may induce an unbalanced Ying and Yang and pathological changes in BCs may be developed. Three main BC types are applied in directing TCM treatments [22]. The Yang-Xu (Yang deficiency) constitution means a decrease in physiological energy and physical function, and certain symptoms may be observed, such as fatigue, shortness of breath, chills, loose stools, and a greater volume of urine [23,24]. The Yin-Xu (Yin deficiency) constitution may be recognized to be insufficient in producing materials such as enzymes to drive body functions; some subjective symptoms may be expressed, such as thirst, hot flush, hard stool, and decreasing urine output. Those symptoms may be experienced by Yin-Xu people [25,26]. When Yin and Yang are uncoordinated for a long time, some symptoms could be observed, such as Qi stagnation, fluid retention, and blood stasis; this would be identified as Stasis. In people with the Stasis BC type, the dynamic of Yin and Yang becomes poorer and body functions gradually decline and become irregular [20,21], and they may experience some subjective symptoms such as tingling pain, chest tightness, and numbness in the limbs. Qi in TCM could be defined as the essential part of life, as it is the fundamental substance or energy to maintain human body activities. Qi flows in certain pathways through the whole body, and this energy could be observed via organ functions or blood flow. When Stasis BC develop, Qi stagnation could be revealed [27].

For TCM physicians, four examinations including inspection, listening, and smelling, inquiry, and palpitation are used to diagnose BCs. However, a body constitution questionnaire (BCQ) is more practical for nurses to use in clinical care. Although there is various research in TCM BCs in chronic diseases, the study in mental health care is insufficient. This study aimed to explore the relationship between TCM BC types and the efficacy of SGAs, and the results may become a bridge when using TCM in a Western medical care context inside hospitals [25].

## 2. Materials and Methods

This study applied a longitudinal research design to investigate the relationship of schizophrenia patients’ BC types and the efficacy of SGAs treatment in the process of acute psychosis treatment. Questionnaire surveys were conducted in the 1st, 3rd, and 6th week after admission to hospital. The trends of psychotic symptom improvements and the trend of BCs type changes under the SGAs treatment were studied.

### 2.1. Study Participants

Patients who were diagnosed with schizophrenia and newly admitted in a mental hospital were approached. Inclusion criteria were: aged over 20 years old and were receiving SGAs treatment; exclusion criteria were: had history of substances abuse.

### 2.2. Data Collection and Measurements

#### 2.2.1. The Body Constitution Questionnaire (BCQ)

The BCQ has 44 items with a 5-point Likert scaling method, and since it has been reported as a reliable and valid instrument for measuring BC types, it was used. In previous studies, Cronbach’s α of BCQ were acceptable, which ranged from 0.55–0.88 in assessing Yang-Xu [24], 0.57–0.85 in assessing Yin-Xu [26], and 0.62–0.88 in assessing Stasis [28]. The intra-class correlation coefficients were greater than 0.7 for most items [23,24,25,26]. Participants were classified into the balanced BC type (gentleness BC type) or unbalanced BC types. The unbalanced BC types are scored as follows: 19 items on Yin-Xu (scores ranging from 19–95, and the cut-off point for the diagnosis of Yin-Xu is a total sub-scale score greater than 29.5) [25], 19 items on Yang-Xu (scores ranging from 19–95, and the cut-off point for the diagnosis of Yang-Xu is a total sub-scale score greater than 30.5) [24], and 16 items on Stasis (scores ranging from 16–80, and the cut-off point for the diagnosis of stasis BC is a total sub-scale score greater than 26.5) [28].

#### 2.2.2. The Positive and Negative Symptom Scale (PANSS)

The PANSS is one of the most widely used instrument of psychiatric symptom severity and treatment efficacy of antipsychotic medications. The PANSS provides a comprehensive assessment of symptomatology and has a high inter-rater reliability and test-retest reliability [29,30]. The PANSS consists of 30 items, and each PANSS item is scored on a 7-point Likert scale, from 1 (absent) to 7 (extreme).

#### 2.2.3. Clinical Global Impression (CGI)

The CGI possesses only 1 item on a 7-point Likert score which aims to rate the symptom severity from 0 (absent) to 6 (extreme) [31,32].

### 2.3. Data Collection

The PANSS and CGI were used to evaluate patients’ psychopathologies on the second day (1st week) of hospitalization and in the 3rd and 6th weeks of hospitalization. All participants completed the BCQ by face-to-face interview at the 6th week of hospitalization. Data were collected by a well-trained senior psychiatric nurse. The participants’ laboratory data and medication information were collected from medical records.

### 2.4. Statistical Analysis

All statistical analyses were performed in SPSS (Statistical Product and Service Solutions) version 20.0. Data were analyzed by using descriptive statistics, t-tests, analysis of variance (ANOVA) and generalized estimating equations (GEE).

### 2.5. Ethic Considerations

This study was approved by the Institutional Review Board of the Ministry of Health and Welfare Tsaotum Psychiatric Center. Research-related information and details of the study process were provided to potential participants, and written informed consent forms were obtained from each participant before the questionnaire survey.

## 3. Results

### 3.1. Demographic Information of Research Participants Subsection

A total of 66 participants were recruited in the study, including 35 participants who were treated with risperidone, 16 with olanzapine, and 15 with clozapine. The mean age was 43.7 years old (SD = 10.0 years old, range from 22–65 years old), and the mean duration of being diagnosed with schizophrenia was 19.9 years (SD = 11.4 years, range from 1–46 years). Table 1, Table 2 and Table 3 compare the basic characteristics and biomarkers of participants with different BC types and treated with different SGAs. Patients with unbalanced BC types showed a higher mean of body mass index (BMI) without statistically significant differences. Most patients had two to three unbalanced BC types simultaneously. Nine out of 15 (60%) clozapine-treated, 12 out of 16 olanzapine-treated (75%) participants and 21 out of 35 (60%) risperidone-treated had unbalanced BC types.

Among patients who were treated with risperidone (*n* = 35), 18 participants (51.4%) were classified with Yin-Xu BC, 14 (40.0%) with Yang-Xu BC, and 13 (37.1%) with Stasis BC. Patients with Stasis BC had significantly higher mean values of haemoglobin and haematocrit than those without Stasis BC type (Table 1). Among patients who were treated with olanzapine (*n* = 16), 9 participants (56.3%) were classified with Yin-Xu BC, 11 (68.8%) with Yang-Xu BC, and 8 (50.0%) with Stasis BC. Patients treated with olanzapine and classified as Yin-Xu BC had significantly higher mean values of blood cholesterol than those who did not possess Yin-Xu BC (Table 2). In patients who were treated with clozapine (*n* = 15), 8 participants (53.3%) had Yin-Xu BC, 8 (53.3%) had Yang-Xu BC, and 6 (40.0%) had Stasis BC. Patients treated with clozapine and classified with Yang-Xu BC exhibited a significantly longer history of schizophrenia and had higher means in perceptions of stress than those who were not classified with Yang-Xu BC (Table 3).

### 3.2. BC Types in Different SGAs Groups

Analysis of variance (ANOVA) was used to compare the scores of 44 items in the BCQs among participants treated with different SGAs. By using the Scheffe post hoc test, there were no significant differences in the mean scores of all 44 items in the BCQs between different SGAs.

### 3.3. Psychopathology Improvements in Different BC Type Groups

Analysis of variance (ANOVA) was used to compare the scores of 44 items in the BCQs among participants treated with different SGAs. By using the Scheffe post hoc test, there were no significant differences in the mean scores of all 44 items in the BCQs between different SGAs. The general estimating equation (GEE) was used to analyze the levels of improvement in CGI and PANSS scores (including positive symptoms sub-score (PANSS-P), negative syndrome sub-score (PANSS-N), general psychopathology sub-score (PANSS-G), and total score (PANSS-T)) between different BC types over the period of 6 weeks of hospitalization. The CGI and PANSS were measured in the 1st (the second day of admission to hospital), the 3rd and the 6th week. The results indicated that there were significant improvements in CGI and PANSS scores over time in all groups (Yin-Xu vs. without Yin-Xu groups; Yang-Xu vs. without Yang-Xu groups; Stasis vs. without Stasis groups), no matter the participants were treated in which kind of SGAs. However, whether in olanzapine, clozapine, or risperidone treatments, there were no significant improvements in group and group–time interactions of CGI and PANSS scores both in the unbalanced BC types and the gentleness BC type (in Table 4 and Table 5; Table 5 only showed PANSS-T).

## 4. Discussion

The symptoms of unbalanced BCs could be difficult to express exactly for schizophrenia patients, and patients treated with olanzapine had a higher prevalence rate in unbalanced BCs. In this era, medicine has transformed from “disease medicine” into “preventive medicine”, and has focused on individual treatment rather than generalize treatment [18]. This study is the first research to explore the relationship between TCM BC types and the efficacy of SGAs in patients with schizophrenia. Over 60% of patients with schizophrenia with SGAs treatment exhibited unbalanced BC types. In addition, a higher prevalence rate of unbalanced BC types in people with schizophrenia was revealed when compared with people with type 2 diabetes (Yin-Xu, Yang-Xu, and Stasis were 27.8, 12.9, and 12.9%, respectively), and people with type 2 diabetes combined with albuminuria (Yin-Xu, Yang-Xu, and Stasis were 30.9, 15.4, and 17.35%, respectively) [22,33]. This result may remind the health care provider to evaluate schizophrenia patients’ health status in a non-invasive method—BC type in questionnaire survey, this evaluation should be an alternative or add-on to an official internationally standardized evaluation on the purpose of preventive care and promoting both physical and mental care quality [34]. The study results reflect individual differences in patients with the same diagnosis [35].

Participants who were treated with olanzapine and possessed the Yin-Xu BC, had blood cholesterol levels significantly higher than those treated with olanzapine without the Yin-Xu BC. In the olanzapine treatment group, those who had unbalanced BC types showed higher levels of fasting blood glucose when compared with those with the gentleness BC type. This result was echoed by previous studies, that SGAs had a greater impact than FGAs on short-term total cholesterol increase, and in the olanzapine group, cholesterol was significantly higher than in the risperidone group [6]. Furthermore, olanzapine was considered to have strong correlations with metabolic problems when compared to other SGAs [1]. A previous pre-clinical model study demonstrated that olanzapine caused significant glucose intolerance and more insulin resistance incidences in the hyperinsulinaemic-euglycemic clamp test; this laboratory method was used as a standardized test in evaluating insulin sensitivity in type 2 diabetes mellitus patients [1,36]. In particular, health providers should evaluate the level of cholesterol, fasting glucose and introduce early interventions to keep appropriate body weight in patients who have olanzapine treatment with unbalanced BC. In patients with unbalanced BCs, their perception of stress was higher than patients with the gentleness BC. In our study, patients with Stasis BC may have worse physical performance, such as greater BMI values (more than 25 kg/m^2^); this finding was consistent with previous research [37]. TCM emphasizes its importance in disease prevention and health promotion. From this study results, providing interventions for patients with schizophrenia and unbalanced BCs to control their body weight, promote their health and prevent hypercholesterolemia may be important. In TCM context, good health can only be attained in physical, mental, social, and spiritual balances.

Schizophrenia patients with Stasis BC should be noticed. Micro-circulation obstruction, abnormal haemorheology, unstable haemodynamics, abnormal platelet and endothelium, dysfunctional anticoagulation, and fibrinolysis: all conditions which lead to blood stasis [38]. Some studies proposed that possessing Stasis BC is significantly associated with hyperlipidaemia, atherosclerosis, coronary heart disease, or coronary artery disease [38,39,40]. Several studies demonstrated that olanzapine is a relatively unsafe drug, especially for patients who have coronary heart disease [41,42,43,44]. In addition, some research showed that antipsychotic drug use is associated with acute myocardial infarction and stroke [45,46,47,48]. People with schizophrenia have shown higher morbidity and mortality rates compared with the general population. From the research results mentioned above, the point of views in Western medicine are consistent with TCM; however, more evidence-based and well-designed experimental research is needed to further confirm these conclusions. Schizophrenia is a complex and multi-factorial mental health problem; long term adherence to medications is necessary for patients in maintaining stability and decreasing symptom recurrence.

Health providers could provide health-promoting interventions according to their unbalanced BC types, particularly in patients who are treated with olanzapine and have a Stasis BC for promoting disease prevention and control [34]. The unbalanced BCs in individuals also represent the susceptibility to chronic diseases, such as diabetic or coronary artery disease [21]. In the long term, TCM may play a more and more important role in preventing metabolic side effects in SGAs users. From the international trend of medical care model, a combination of Western medical treatment and complementary/alternative therapies are developing; understanding how TCM BCs influence the body dynamics could be the first step to establishing communication among various treatments [18].

A significant improvement in the CGI and PANSS scores was observed during the 6th week in all treatment groups across time. However, there were no statistically significant group–time interactions in the CGI and PANSS scores decreases in groups with or without Yin-Xu, Yang-Xu, and Stasis BCs under different SGA treatments. Theoretically, the effects of prescribed medicine are related to personal constitution, and the efficiency of prescribed medicine can be improved according to the patient’s BC types [49]. However, the relationship between BC types and the efficacy of SGAs did not reach the statistical significance. This may be due to the small sample size in this study and that the data collection was limited in central Taiwan, which may diminish the generalizability of the results. Besides, since the follow-up period of this study was only 6 weeks, the survey period may be too short to comprehensively explore the changes of SGAs efficacy and BCs. One limitation of this study is a small sample size with 66 participants, nevertheless, the research results should be considered as a valid contribution in the TCM field with three time points follow-up for evaluating the BC changes of such a specific population. In future studies, a longer follow-up period and a larger sample size are suggested. Some factors which may affect BCs, such as living environment, should be measured, and whether the BCs deviation would influence the recurrence of disease could be investigated [50]. In addition, the relationship between health interventions (e.g., exercise, diet) and the transformation of BCs is worthwhile to look at in further investigations. BCs in TCM has the potential to be used as an evaluation tool for planning health promotion strategies in mental care.

## 5. Conclusions

This is the first study to explore the longitudinal relationship between BC types and the efficacy of SGAs in schizophrenia patients. Although there were no significant changes in the group and group–time interaction in regard to CGI and PANSS scores improvements between unbalanced BCs patients and gentleness BC patients, BC types evaluations in aging mental disease patients is still suggested for health evaluation. According to the holistic care concept of TCM and Western medicine, maintaining and promoting health may be achieved through understanding an individual’s BC types and keeping balanced BCs. Those patients who were treated with olanzapine had a higher proportion in overweight problems, and healthcare providers should pay more attention if their BC types show to be unbalanced, which may be related to some prevalence of chronic diseases. This research results suggested that a BC type evaluation could be a good alternative assessment tool in schizophrenia patients in monitoring their health status. Furthermore, patients who are taking olanzapine, especially those who have an unbalanced BC type may need to monitor the levels of cholesterol and fasting glucose. This may be helpful in preventing health problems and chronic diseases in aging psychological patients. The result of this research offers empirical evidence to the health care providers in better understanding the use of TCM BCs to evaluate patients’ health status in psychology care and to be the base of intervention-based research.

## Figures and Tables

**Table 1 healthcare-09-01480-t001:** Comparisons of participants’ characteristics and biomarkers between constitutions of Yin-Xu, Yang-Xu, and Stasis (treated with risperidone, *n* = 35).

Variable	Yin-Xu			Yang-Xu			Stasis		
	No (*n* = 17)	Yes (*n* = 18)	*p* Value	No (*n* = 21)	Yes (*n* = 14)	*p* Value	No (*n* = 22)	Yes (*n* = 13)	*p* Value
Age (years)	44.7 ± 11.4	43.0 ± 11.0	0.644	43.6 ± 11.0	44.1 ± 11.7	0.913	44.0 ± 10.8	43.5 ± 12.0	0.907
History of schizophrenia (years)	18.3 ± 12.8	17.8 ± 12.1	0.901	18.2 ± 11.9	17.8 ± 13.3	0.919	17.6 ± 11.4	18.8 ± 14.1	0.771
BMI (kg/m^2^)	25.0 ± 5.6	24.1 ± 5.0	0.634	24.5 ± 5.3	24.6 ± 5.3	0.982	23.8 ± 5.2	25.8 ± 5.3	0.278
Perception of stress (0–10)	2.0 ± 2.4	4.61 ± 3.77	0.02 *	3.1 ± 3.4	3.6 ± 3.5	0.676	3.1 ± 3.4	3.9 ± 3.5	0.508
Perception of health (0–100)	82.6 ± 15.1	81.2 ± 12.0	0.857	82.6 ± 14.0	80.1 ± 12.8	0.599	81.6 ± 13.8	81.7 ± 13.2	0.983
Fasting glucose (mg/dL)	97.8 ± 32.5	91.4 ± 17.0	0.478	94.6 ± 30.0	94.4 ± 18.0	0.981	90.5 ± 18.5	101.3 ± 34.3	0.230
Triglycerides (mg/dL)	105.8 ± 54.2	117.7 ± 61.4	0.549	98.7 ± 47.0	131.8 ± 67.3	0.124	103.5 ± 52.3	126.2 ± 65.0	0.266
Cholesterol (mg/dL)	173.1 ± 36.0	168.9 ± 36.6.	0.736	171.3 ± 35.0	170.4 ± 38.4	0.943	176.7 ± 36.6	161.2 ± 33.5	0.221
Haemoglobin	13.3 ± 1.4	13.5 ± 1.3	0.752	13.3 ± 1.6	13.6 ± 0.8	0.500	13.0 ± 1.3	14.2 ± 1.1	0.01 **
Haematocrit	40.1 ± 3.5	40.2 ± 3.5	0.914	39.8 ± 4.0	40.7 ± 2.3	0.367	39.1 ± 3.5	42.0 ± 2.6	0.013 **

Data were presented as the mean ± SD; * *p* < 0.05, ** *p* < 0.01. BMI: body mass index.

**Table 2 healthcare-09-01480-t002:** Comparisons of participants’ characteristics and biomarkers between constitutions of Yin-Xu, Yang-Xu, and Stasis (treated with olanzapine, *n* = 16).

Variable	Yin-Xu			Yang-Xu			Stasis		
	No (*n* = 7)	Yes (*n* = 9)	*p* Value	No (*n* = 5)	Yes (*n* = 11)	*p* Value	No (*n* = 8)	Yes (*n* = 8)	*p* Value
Age (years)	41.5 ± 10.0	45.9 ± 6.0	0.327	38.3 ± 10.3	46.5 ± 5.5	0.154	43.5 ± 8.4	44.4 ± 8.2	0.824
History of schizophrenia (years)	15.5 ± 7.0	21.0 ± 11.0	0.259	12.7 ± 5.0	21.2 ± 9.7	0.91	13.5 ± 6.2	23.6 ± 9.4	0.024 *
BMI (kg/m^2^)	23.6 ± 7.0	24.2 ± 4.9	0.851	24.7 ± 8.0	23.5 ± 4.8	0.716	22.5 ± 4.9	25.3 ± 6.4	0.344
Perception of stress (0–10)	3.0 ± 2.9	6.6 ± 2.4	0.019 *	2.0 ± 2.4	6.36 ± 2.5	0.005 **	4.4 ± 3.81	5.6 ± 2.4	0.445
Perception of health (0–100)	63.6 ± 20.1	57.9 ± 26.9	0.649	67.0 ± 15.7	57.4 ± 26.6	0.468	67.0 ± 26.9	53.6 ± 19.2	0.276
Fasting glucose (mg/dL)	91.0 ± 17.4	96.6 ± 24.0	0.614	89.8 ± 21.1	96.1 ± 21.5	0.594	92.4 ± 17.0	95.9 ± 25.2	0.75
Triglycerides (mg/dL)	100.7 ± 41.5	160.4 ± 108.9	0.193	100.0 ± 39.3	149.9 ± 102.2	0.316	135.4 ± 101.1	133.3 ± 82.5	0.964
Cholesterol (mg/dL)	171.6 ± 23.5	206.9 ± 36.3	0.043 *	176.0 ± 22.1	198.5 ± 38.8	0.252	191.6 ± 39.0	191.3 ± 34.0	0.984
Haemoglobin	13.2 ± 1.3	13.9 ± 1.7	0.364	13.2 ± 1.60	13.8 ± 1.6	0.478	14.2 ± 1.6	13.0 ± 14	0.112
Haematocrit	39.6 ± 4.0	42.3 ± 5.1	0.263	39.6 ± 4.9	41.8 ± 4.7	0.403	43.0 ± 4.7	39.3 ± 4.18	0.119

Data were presented as the mean ± SD; * *p* < 0.05, ** *p* < 0.01. BMI: body mass index.

**Table 3 healthcare-09-01480-t003:** Comparisons of participants’ characteristics and biomarkers between constitutions of Yin-Xu, Yang-Xu, and Stasis (treated with clozapine, *n* = 15).

Variable	Yin-Xu			Yang-Xu			Stasis		
	No (*n* = 7)	Yes (*n* = 8)	*p* Value	No (*n* = 7)	Yes (*n* = 8)	*p* Value	No (*n* = 9)	Yes (*n* = 6)	*p* Value
Age (years)	39.2 ± 8.4	46.6 ± 10.3	0.16	38.8 ± 7.4	47.0 ± 10	0.112	42.9 ± 10.5	43.5 ± 10.0	0.920
History of schizophrenia (years)	22.5 ± 7.4	28.3 ± 12.0	0.29	19.1 ± 4.9	31.3 ± 10.4	0.014 *	24.0 ± 10.9	28.0 ± 9.4	0.481
BMI (kg/m^2^)	21.6 ± 3.1	24.3 ± 5.9	0.308	21.0 ± 2.6	24.9 ± 5.8	0.129	21.4 ± 2.71	25.5 ± 6.5	0.189
Perception of stress (0–10)	2.0 ± 2.6	3.9 ± 3.4	0.256	1.3 ± 2.2	4.5 ± 3.1	0.039 *	2.3 ± 2.9	4.0 ± 3.4	0.325
Perception of health (0–100)	72.9 ± 36.8	61.3 ± 24.7	0.481	80.0 ± 22.4	55.0 ± 33.0	0.115	71.1 ± 33.0	60.0 ± 27.6	0.508
Fasting glucose (mg/dL)	97.7 ± 20.0	97.3 ± 12.4	0.957	102.9 ± 18.1	92.8 ± 12.8	0.228	99.6 ± 18.0	94.3 ± 12.6	0.550
Triglycerides (mg/dL)	113.7 ± 127.4	105.4 ± 69.1	0.875	70.1 ± 20.5	143.5 ± 124.5	0.142	104.1 ± 112.7	117.0 ± 76.1	0.811
Cholesterol (mg/dL)	159.7 ± 28.9	155.5 ± 33.6	0.8	159.3 ± 27.9	155.9 ± 34.4	0.838	168.3 ± 31.9	141.2 ± 21.1	0.068
Haemoglobin	13.1 ± 0.6	13.6 ± 1.4	0.385	12.8 ± 0.6	13.9 ± 1.3	0.057	13.1 ± 0.7	13.9 ± 1.5	0.159
Haematocrit	39.6 ± 1.3	41.1 ± 4.1	0.358	38.9 ± 1.8	41.7 ± 3.5	0.081	39.3 ± 1.7	42.1 ± 4.0	0.081

BMI: body mass index.

**Table 4 healthcare-09-01480-t004:** General estimating equation for the comparison of CGI between different BC types within each SGA treatment group within the study’s time frame.

Variables	Time			Group × Time		
	Beta	SE	*p* Value	Beta	SE	*p* Value
**Risperidone (*n* = 35)**						
Yin-Xu vs. without Yin-Xu	−0.735	0.734	0.000 **	0.096	0.106	0.364
Yang-Xu vs. without Yang-Xu	−0.738	0.641	0.000 **	0.131	0.111	0.236
Stasis vs. without Stasis	−0.705	0.615	0.000 **	0.051	0.117	0.666
**Olanzapine (*n* = 16)**						
Yin-Xu vs. without Yin-Xu	−0.929	0.187	0.000 **	0.04	0.242	0.869
Yang-Xu vs. without Yang-Xu	−0.90	0.261	0.001 **	−0.009	0.289	0.975
Stasis vs. without Stasis	−0.812	0.175	0.000 **	−0.187	0.233	0.421
**Clozapine (*n* = 15)**						
Yin-Xu vs. without Yin-Xu	−0.571	0.157	0.000 **	−0.054	0.249	0.830

SGA treatment group within the study’s time frame. ** *p* < 0.01.

**Table 5 healthcare-09-01480-t005:** General estimating equation for comparison of PANSS-T between different BC types within each SGA treatment group within the study’s time frame.

Variables	Time			Group × Time		
	Beta	SE	*p* Value	Beta	SE	*p* Value
**Risperidone (*n* = 35)**						
Yin-Xu vs. without Yin-Xu	−17.088	1.917	0.000 **	1.005	2.419	0.678
Yang-Xu vs. without Yang-Xu	−17.595	1.639	0.000 **	2.56	2.329	0.272
Stasis vs. without Stasis	−16.886	1.627	0.000 **	0.848	2.354	0.719
**Olanzapine (*n* = 16)**						
Yin-Xu vs. without Yin-Xu	−17.429	3.255	0.000 **	−2.294	3.923	0.559
Yang-Xu vs. without Yang-Xu	−16.9	4.442	0.000 **	−2.645	4.809	0.582
Stasis vs. without Stasis	−16.562	3.008	0.000 **	−4.312	3.653	0.238
**Clozapine (*n* = 15)**						
Yin-Xu vs. without Yin-Xu	−14.14	4.608	0.002 **	−2.295	6.107	0.707
Yang-Xu vs. without Yang-Xu	−15.930	4.888	0.001 **	1.054	6.178	0.865
Stasis vs. without Stasis	−16.667	3.917	0.000 **	3.250	6.146	0.597

** *p* < 0.01.

## Data Availability

The study is available from the 12th months and ended in the 36th months from the article has been published. Please contact the corresponding author, J-F C, with reasonable request reason by e-mail feny@mail.cmu.edu.tw.

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
