# Peer review of "The Relationship between the Second-Generation Antipsychotics Efficacy and the Traditional Chinese Medicine Body Constitutions in Patients with Schizophrenia"

_healthcare, 2021, doi:10.3390/healthcare9111480_

Round 1
Reviewer 1 Report
I really appreciate the opportunity to review the manuscript healthcare-1391132 entitled:
"The Relationship between the Second-generation Antipsychotics Efficacy and the Traditional Chinese Medicine Body Constitutions in Patients with Schizophrenia"
I commend the authors for describing this critical and timely issue. The paper is interesting and well written; however, I would like to highlight some issues that merit revision:
Line 72: "such as qi stagnation" nowhere in the manuscript is it explained what is scientifically meant by Qi. Please, add a clear definition.
Line 73: "body regulation is less efficient", from the text it seems a general dysregulation. I ask the authors to clarify in the text if this means a kind of Multi-Organ Disease Failure.
Results: "and lower mean of perception of 139 health." I beg the authors to describe this in more detail.
Line 205. "This result may remind the health care provider to evaluate schizophrenia patients' health status in a non-invasive method - BC type evaluation." Please provide solid evidence to support this hypothesis; in addition, it seems appropriate to clarify whether such a non-invasive evaluation should be an alternative or add-on to an official internationally standardized evaluation.
"Considering the difficulty of data collection in schizophrenia patients" this statement is not appropriate for a journal such as Healthcare, please remove the sentence.
Limitations are not specified at any point in the text; although some weaknesses in the article, e.g., "Sample size is only 66" it seems appropriate to indicate them accompanied by an introductory sentence such as "The limitations of this study are"
Manuscript lines 47b through 52 are redundant, please consider reducing the text considerably.
Furthermore, I suggest removing references tending to excessive promotion of the founding theory of the article (the basics of TCM), I believe that the purpose of an article is to provide the reader with the elements to create their own opinion.
At the moment, to the best of my knowledge, I do not recognize a robust scientificity of the concepts yin-xu and yang-xu, which are proposed in the article; for this reason, I refer to the editor's decision.
Author Response
Thank you for the crucial suggestions to our manuscript. Please see the attached to view what has been revised. Thanks again.

Reviewer 2 Report
-- The paper is an account of relationship between body composition based on traditional Chinese medicine and efficacy of three antipsychotics
-- There are numerous typos and grammar errors, e.g. the very first sentence the word "severe" is written as "sever"
-- Introduction could use more references
-- The description of design as "longitudinal research method" needs specification with rationale
-- The data analysis is appropriate
-- However, the results of the study do not add anything substantive to the literature. The significance to the field is not well established
-- The conclusions extend beyond the data. These need to be limited to significant findings
Author Response

(The authors gave the same response as above.)

Round 2
Reviewer 1 Report
The paper is very interesting and well written, methodologically unexceptionable, and the new implementations provide a valid contribution to the work. Every requested correction has been done, and the manuscript is now suitable for publication
Reviewer 2 Report
The manuscript needs English language editing and with that it can be published.